# Automated multi-class classification for prediction of tympanic membrane changes with deep learning models

Yeonjoo Choi[1‡], Jihye Chae[2‡], Keunwoo Park[2], Jaehee Hur[2], Jihoon Kweon[2]*, Joong Ho Ahn[1]*

1 Department of Otorhinolaryngology-Head and Neck Surgery, University of Ulsan College of Medicine, Asan Medical Center, Seoul, Korea, 2 Departments of Convergence Medicine, University of Ulsan College of Medicine, Asan Medical Center, Seoul, Korea

‡ These authors contributed equally to this work and share first authorship on this work.
* meniere@amc.seoul.kr (JHA); kjihoon2@naver.com (JK)

## Abstract

### Backgrounds and objective

Evaluating the tympanic membrane (TM) using an otoendoscope is the first and most important step in various clinical fields. Unfortunately, most lesions of TM have more than one diagnostic name. Therefore, we built a database of otoendoscopic images with multiple diseases and investigated the impact of concurrent diseases on the classification performance of deep learning networks.

### Study design

This retrospective study investigated the impact of concurrent diseases in the tympanic membrane on diagnostic performance using multi-class classification. A customized architecture of EfficientNet-B4 was introduced to predict the primary class (otitis media with effusion (OME), chronic otitis media (COM), and 'None' without OME and COM) and secondary classes (attic cholesteatoma, myringitis, otomycosis, and ventilating tube).

### Results

Deep-learning classifications accurately predicted the primary class with dice similarity coefficient (DSC) of 95.19%, while misidentification between COM and OME rarely occurred. Among the secondary classes, the diagnosis of attic cholesteatoma and myringitis achieved a DSC of 88.37% and 88.28%, respectively. Although concurrent diseases hampered the prediction performance, there was only a 0.44% probability of inaccurately predicting two or more secondary classes (29/6,630). The inference time per image was 2.594 ms on average.

**Data Availability Statement:** The datasets generated and/or analyzed during the current study are not publicly available because permission of

sharing patient data was not granted by the Institutional Review Board (IRB) of the Asan Medical Center (AMC). In order to access the dataset, it is required to submit a request and obtain an approval from the IRB of AMC (E-mail: irb@amc.seoul.kr; Tel: +82-2-3010-7172).

**Funding:** This work was supported by the National Research Foundation of Korea(NRF) grant funded by the Korea government(MSIT).(No. 2021R1A2C2010048). The funders had no role in study design, data collection and analysis, decision to publish, or preparation of the manuscript.

**Competing interests:** The authors have declared that no competing interests exist.

## Conclusion

Deep-learning classification can be used to support clinical decision-making by accurately and reproducibly predicting tympanic membrane changes in real time, even in the presence of multiple concurrent diseases.

## Introduction

In the otologic field, evaluating the tympanic membrane (TM) and the middle ear via endoscopic evaluation is usually the first step for patients complaining of earache or other problem such as hearing loss, dizziness, or facial palsy [1]. To evaluate otologic diseases such as acute/chronic otitis externa or acute/chronic otitis media, it is important to examine the state of the external auditory canal (EAC) and TM using common tools like the otoscope, which allows for simple observation and diagnosis. Apart from being a primary diagnostic step, an accurate otoscopic exam can also guide the correct course of treatment during the follow up period. Given how important it is to diagnose and evaluate accurately the state of disease during the follow up period, intensive training is required before being able to accurately diagnose the condition [1]. Unfortunately, misdiagnosis in the clinical field is still fairly common.

One study reported that diagnostic accuracy varied among physicians, including otolaryngologists, pediatricians, and family medicine doctors [2]. Another study reported that otolaryngologists diagnosed these otologic diseases with 73% accuracy while pediatricians and general practitioners had an accuracy rate of 50% and 64%, respectively [3]. Therefore, even though there is a glaring need for trained otolaryngologists to make accurate diagnoses, the limited number of specialists makes it impossible [4]. Therefore, there is a need to develop a modality that can accurately evaluate the status of EAC and TM to support the diagnostic system. Specifically, there is a need for an image-based diagnostic algorithm based on otoscopic images.

In recent years, advances in image classification using deep learning networks have been proven to improve the diagnosis performance of middle ear diseases [5–8]. Khan et al. [9] reported that classification accuracy of deep network reached 94.9% in the classification of normal, chronic otitis media (COM) with TM perforation, and otitis media with effusion (OME). Detection of tympanic perforation had an accuracy rate of 91% [10]. The ensemble approach, which combines the outputs of multiple networks, enhanced predictability in the categorical classification of otoendoscopic images [11,12]. Deep learning prediction can help clinicians make more accurate decisions [13]. Although previous studies showed the potential applicability of deep learning-based diagnosis, otoendoscopic images of multiple diseases that could hamper diagnostic accuracy were excluded from the prediction.

Therefore, in this study, we built a database of otoendoscopic images containing multiple diseases to investigate the impact of concurrent diseases on the classification performance of deep learning networks.

## Materials and methods

### Data description

Otoendoscopic images of TM were collected from patients who visited the otologic clinic in Asan Medical Center from Jan 2018 to Dec 2020. In clinical practice, the otoendoscopic video sequence was taken for diagnostic examination and an image frame visualizing the whole TM was stored in the hospital system without patient-identifiable information. Otoendoscopic

images enrolled based on the date of visit were completely anonymized before being provided by the hospital system. The collected images were classified into one primary class and four secondary classes according to their diagnostic classification. The categories of each image were blindly annotated by two otologists with 26 and 5 years of experience, respectively. A total of 6,630 otoendoscopic images labeled identically by two annotators were included in this study. The primary class was annotated as one of otitis media with effusion (OME, 1,630 images), chronic otitis media (COM, 1,534 images), and 'None' (3,466 images)–meaning the absence of OME and COM. OME refers to effusions in the middle ear cavity, which manifest in the air-fluid level or as an amber-like color change of TMs. COM refers to a perforated TM. Binary labels were given for the secondary classes of attic cholesteatoma (893 images), myringitis (1,083 images), otomycosis (181 images), and ventilating tube (1,676 images) (Fig 1). Attic cholesteatoma refers to any sign of retraction pocket in attic or visible attic destruction. Myringitis is defined as any inflammation of the tympanic membrane, including acute otitis media. Otomycosis refers to a fibrinous accumulation of debris or visible pores of fungus in the external auditory canal. Ventilating tube refers to an inserted tube across the TM. For example, when a TM was normal, the primary class was 'None' and the secondary classes were 'False' for attic cholesteatoma, myringitis, otomycosis, and ventilating tube (Fig 2B). An otoendoscopic image with only otomycosis was assigned 'None' for the primary class, 'True' for otomycosis, and 'False' for the other secondary classes. For 3,508 images, one or more secondary classes were positive. The present study is in compliance with the Declaration of Helsinki and research approval was granted from the Institutional Review Board of the Asan Medical Center with a waiver of research consent (IRB no. 2021–0837).

## Deep learning network

The architecture of EfficientNet-B4 [14] was customized to have shared and task-specific layers for the multi-task learning (Fig 2A). The task-specific layers consisted of five shallow classifiers corresponding to the primary class and four secondary classes ('combined model'). Parameters between the classifiers were not shared.

As an input to deep networks, RGB images reformatted into 256×256×3 with circular cropping were used (Fig 2A). Data augmentation was performed by randomly applying rotation (−90˚ to 90˚), translation shift (0–20% of image size in horizontal and vertical axes), zoom (0–20%), horizontal flip, brightness change (0–20%) and downscale (0–50%). The pre-trained weight from ImageNet was applied for transfer learning. Categorical cross-entropy loss was adopted to train the models for multi-class classification, which is defined as,

$$L_{CCE} = -\frac{1}{N}\sum_{i}^{N}\sum_{c}^{M} t_{i,c} log(p_{i,c})$$

where $N$ is the number of training samples, $M$ is the number of classes, $t_{i,c}$ is the ground truth, and $p_{i,c}$ is the output probability. The final output was determined as the primary rank of the softmax value.

## Training setup and evaluation metrics

The deep learning model implemented using Pytorch was trained on a workstation with AMD Ryzen 7 5800X CPU 3.8 GHz, 128 GB RAM, and two NVIDIA Geforce RTX 3090 Ti GPUs. The model training was conducted for 200 epochs at maximum with a mini-batch size of 32. For training, an Adam optimizer was applied with β1 = 0.9 and β2 = 0.9999. The learning rate was initially set as $10^{-3}$ and was reduced by half with a saturation criteria of 50 epochs.

| | Primary class | None | OME | COM |
|---|---|---|---|---|
| Secondary class | | n = 3,466 | n = 1,630 | n = 1,534 |
| No secondary class<br><br>n = 3,122 | | <br>n = 692 | <br>n = 1,433 | <br>n = 997 |
| Attic Cholesteatoma<br><br>n = 744 | | <br>n = 610 | <br>n = 95 | <br>n = 39 |
| Myringitis<br><br>n = 842 | | <br>n = 439 | <br>n = 17 | <br>n = 386 |
| Otomycosis<br><br>n = 58 | | <br>n = 18 | <br>n = 3 | <br>n = 37 |
| Ventilating tube<br><br>n = 1,546 | | <br>n = 1,468 | <br>n = 74 | <br>n = 4 |
| Multiple secondary classes<br><br>n = 318 | | <br>n = 239 | <br>n = 8 | <br>n = 71 |

**Fig 1. Classification of otoendoscopic images by primary and secondary classes with representative examples.**
OME, otitis media with effusion; COM, chronic otitis media.

The evaluation metrics for each label were precision, sensitivity (recall), specificity, and dice similarity coefficient (DSC), which were defined as precision = TP / (TP + FP), sensitivity = TP / (TP + FN), specificity = TN / (FP + TN) and DSC = 2 × precision × recall / (precision + recall), where TP is true positive, FP is false positive, and FN is false negative. The per-class accuracy was calculated by dividing the sum of TPs and TNs with the total number of images in a fold.

For 5-fold cross validation, the dataset was divided so that each fold contained an equal number of images (n = 1,326). The fold proportion of training, validation, and test sets was fixed at 3:1:1 and their compositions were changed under cyclic permutation.

## Separate prediction for single class as reference

To evaluate the performance of multi-class classification, the deep learning models for the prediction of each class were separately trained ('separate model'). In this setting, only one classifier for the target class remained in the task-specific layers (Fig 2A).

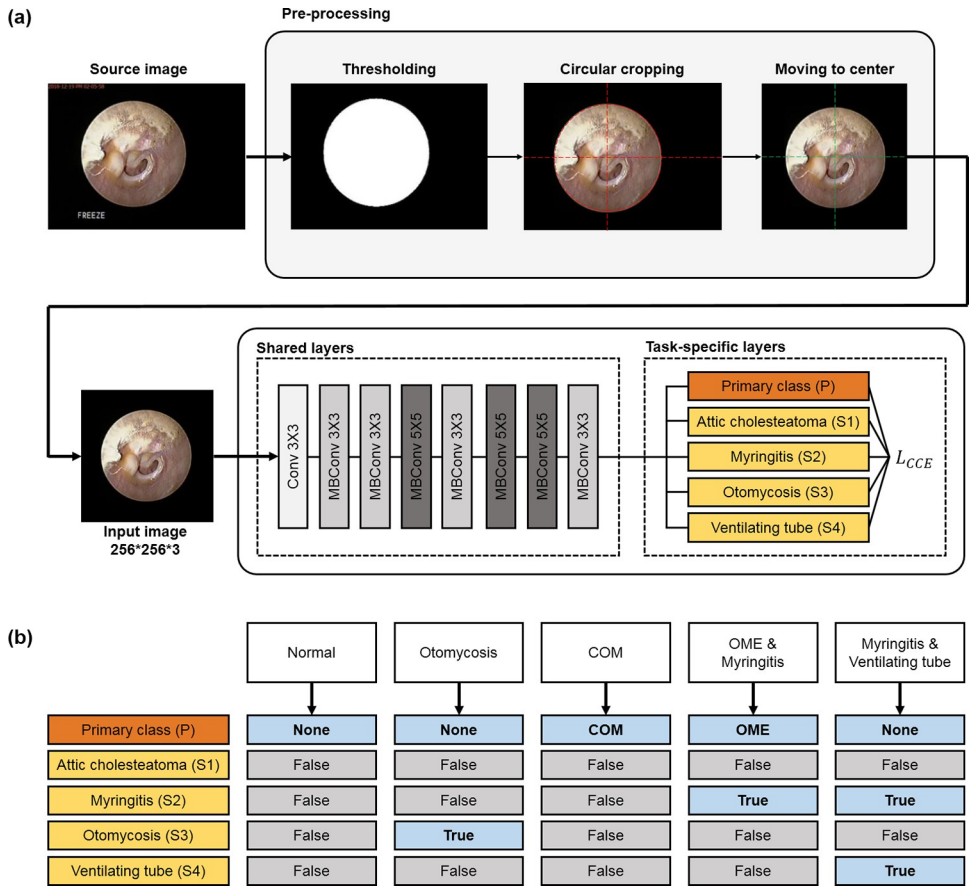

**Fig 2.** (a) Schematic diagram of deep learning network for multi-class classification of otoendoscopic images. (b) Labeling examples. For a normal tympanic membrane (TM), the otoendoscopic image was labeled as 'None' for the primary class and 'False' for the secondary classes (attic cholesteatoma, myringitis, otomycosis and ventilating tube). When TM was diseased as one of the secondary classes without otitis media with effusion (OME) and chronic otitis media (COM), the primary class was given as 'None' for the otoendoscopic image.

**Table 1. Prediction performance of combined model for primary and secondary classes.** McNemar test was applied for the comparison with separate models, denoted with the subscript 'sep'.

| | DSC | Accuracy | Sensitivity | Precision | Specificity | DSC$_{sep}$ | DSC—DSC$_{sep}$ | p-value |
|---|---|---|---|---|---|---|---|---|
| Primary class (P) | 95.19% | 95.32% | 95.38% | 95.32% | 94.65% | 94.90% | 0.29% | 0.360 |
| None | 95.68% | - | 96.91% | 94.49% | 93.58% | 95.49% | 0.19% | - |
| OME | 93.80% | - | 91.90% | 95.78% | 96.44% | 93.76% | 0.04% | - |
| COM | 96.09% | - | 95.37% | 96.82% | 95.31% | 95.46% | 0.63% | - |
| Attic cholesteatoma (S1) | 88.37% | 96.97% | 85.54% | 91.39% | 98.74% | 87.75% | 0.62% | 0.663 |
| Myringitis (S2) | 88.28% | 96.21% | 87.26% | 89.32% | 97.96% | 88.58% | -0.30% | 0.404 |
| Otomycosis (S3) | 72.38% | 98.69% | 62.98% | 85.07% | 99.69% | 68.26% | 4.12% | 0.030 |
| Ventilating tube (S4) | 98.89% | 99.44% | 98.57% | 99.22% | 99.74% | 98.68% | 0.21% | 0.879 |

## Statistical analysis

Categorical variables are presented as numbers and percentages. The McNemar test was applied to compare DSC values between combined and separate models. Statistical analyses were performed using R package.

## Results

### Classification performance of combined model

In the prediction of the primary class, the overall dice similarity coefficient (DSC) was 95.19%, with COM achieving the highest DSC of 96.09% (Table 1). Misidentification between COM and OME rarely occurred (7 images), and most of the prediction errors appeared as false positives and false negatives in the 'None' class (Fig 3). Among the secondary classes, the

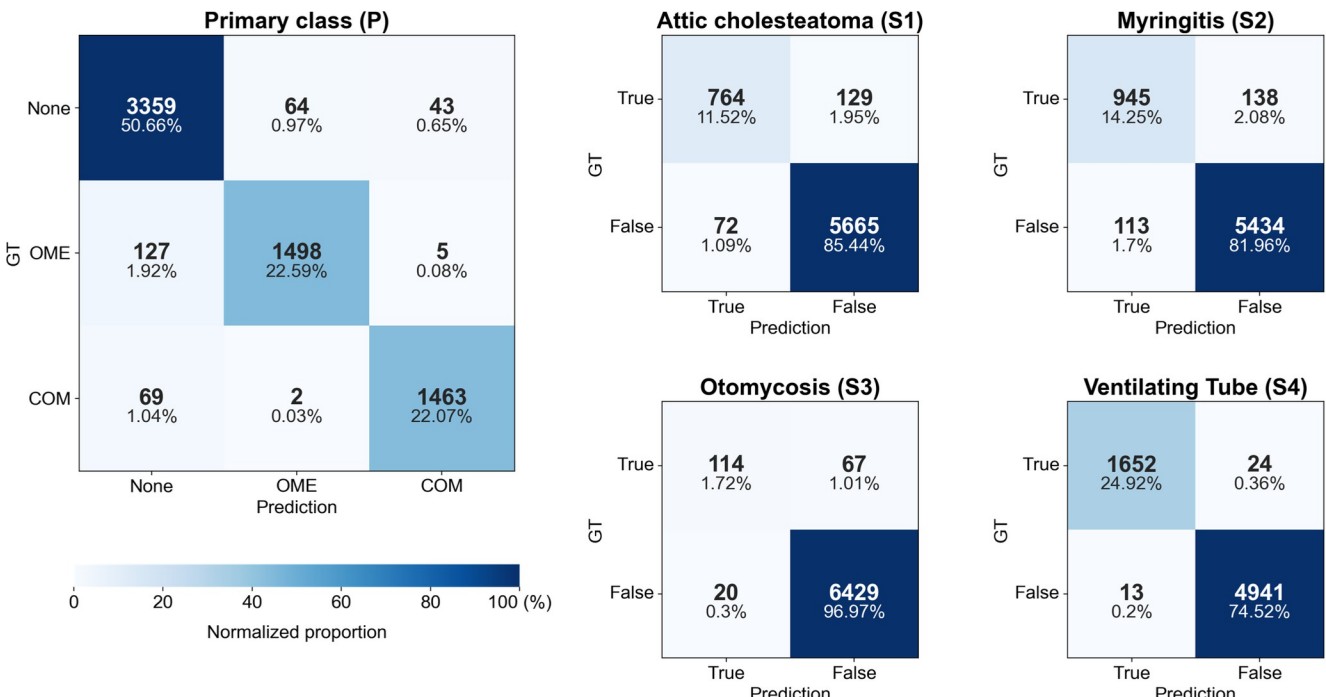

**Fig 3. Confusion matrix of combined model in 5-fold cross validation for the prediction of primary and secondary classes.** GT, ground truth; OME, otitis media with effusion; COM, chronic otitis media.

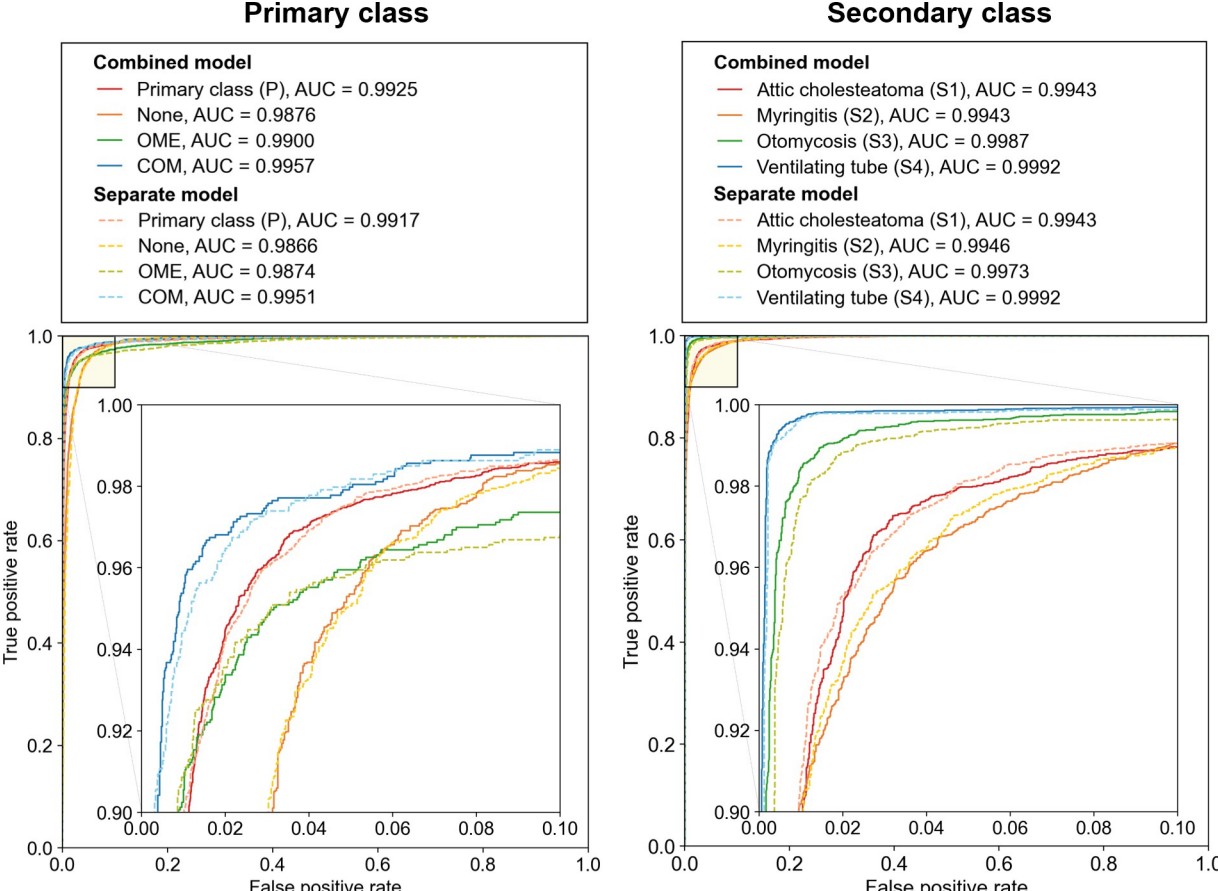

**Fig 4. Receiver operating characteristics (ROC) curves and AUC values for primary and secondary classes.** Micro-average was applied to evaluate the overall predictability of deep learning model for the primary class. AUC, area under the ROC curve; OME, otitis media with effusion; COM, chronic otitis media.

ventilating tube was most accurately diagnosed (DSC = 98.89%), followed by attic cholesteatoma and myringitis with DSCs of 88% or higher (Table 1). Otomycosis, which trained with fewer positive cases, had lower predictive accuracy than other classes. The AUC values for the primary and secondary classes were ≥ 0.9925 (Fig 4).

## Impact of concurrent diseases

With a greater number of positive secondary classes, the probability of accurate prediction for all classes gradually decreased from 92.57% to 14.29% (Table 2). When the number of positives in the secondary classes ≥ 2, the proportion of images with at least one false prediction was over 40%. Nonetheless, the combined model had only a 0.44% probability of inaccurately predicting two or more secondary classes (29/6,630).

## Comparison with separate models

Compared to the separate models, the combined model slightly improved the predictability of the deep learning models except for myringitis, albeit not in a statistically significant way (Table 1). The combined model provided correct diagnoses for all classes in 88.1% of the images (5,841/6,630), which was 0.98% higher than the separate models (Table 2, p = 0.009).

**Table 2. Comparison of prediction accuracy between combined and separate models according to the number of positives in the secondary classes.**

| | Number of positives in the secondary classes | Primary class correct | | | | | Primary class incorrect | | | | |
|---|---|---|---|---|---|---|---|---|---|---|---|
| | | Number of the secondary classes incorrectly predicted | | | | | Number of the secondary classes incorrectly predicted | | | | |
| | | 4 | 3 | 2 | 1 | 0 | 4 | 3 | 2 | 1 | 0 |
| Combined model | 0 (n = 3,122) | - | - | - | 127 (4.07%) | 2,890 (92.57%) | - | - | - | 14 (0.45%) | 91 (2.91%) |
| | 1 (n = 3,190) | - | 1 (0.03%) | 8 (0.25%) | 222 (6.96%) | 2,777 (87.05%) | - | - | 3 (0.09%) | 33 (1.03%) | 146 (4.58%) |
| | 2 (n = 311) | - | 1 (0.32%) | 10 (3.22%) | 104 (33.44%) | 173 (55.63%) | - | 1 (0.32%) | 2 (0.64%) | 12 (3.86%) | 8 (2.57%) |
| | 3 (n = 7) | - | - | 3 (42.86%) | 3 (42.86%) | 1 (14.29%) | - | - | - | - | - |
| | Sum (n = 6,630) | - | 2 (0.03%) | 21 (0.32%) | 456 (6.88%) | 5,841 (88.10%) | - | 1 (0.02%) | 5 (0.08%) | 59 (0.89%) | 245 (3.70%) |
| Separate model | 0 (n = 3,122) | - | - | 4 (0.13%) | 122 (3.91%) | 2,857 (91.51%) | - | - | - | 16 (0.51%) | 123 (3.94%) |
| | 1 (n = 3,190) | - | - | 13 (0.41%) | 269 (8.43%) | 2,743 (85.99%) | - | - | 1 (0.03%) | 16 (0.50%) | 148 (4.64%) |
| | 2 (n = 311) | - | - | 10 (3.22%) | 106 (34.08%) | 172 (55.31%) | - | 1 (0.32%) | - | 14 (4.50%) | 8 (2.57%) |
| | 3 (n = 7) | - | - | 2 (28.57%) | 1 (14.29%) | 4 (57.14%) | - | - | - | - | - |
| | Sum (n = 6,630) | - | - | 29 (0.44%) | 498 (7.51%) | 5,776 (87.12%) | - | 1 (0.02%) | 1 (0.02%) | 46 (0.69%) | 279 (4.21%) |

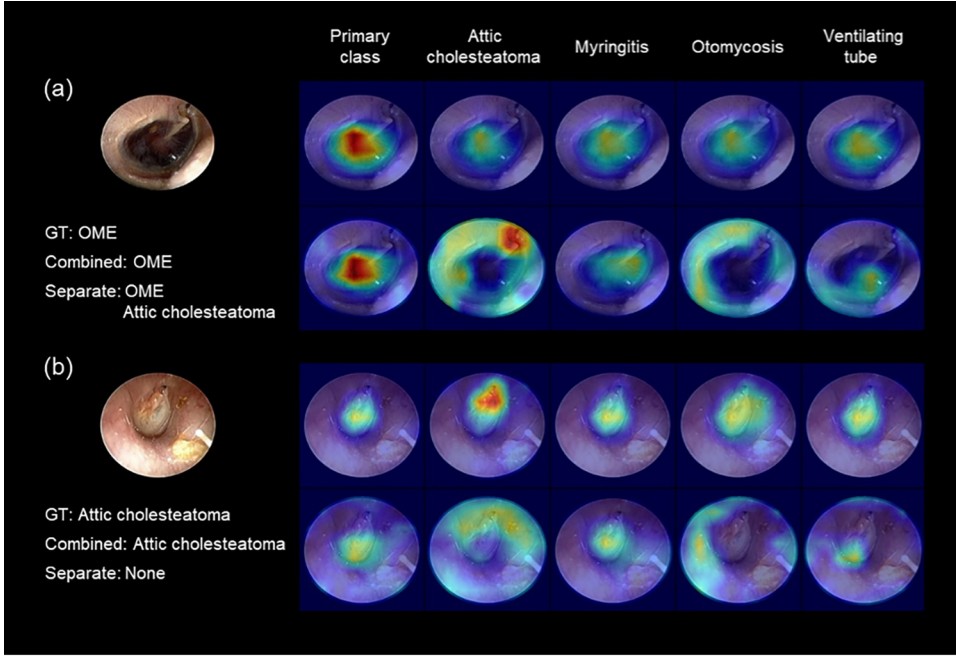

**Fig 5.** Grad-CAM visualization of representative examples for combined (upper row) and separate (lower row) models. The red area refers to the part of the model where the attention is strong. GT, ground truth; OME, otitis media with effusion.

**Table 3. Computational cost and inference time for application of deep-learning classification for tympanic membrane changes.**

| Model | | Number of parameters (million) | Training time (s) | Training time per epoch (s) | Inference time per image (ms) |
|---|---|---|---|---|---|
| Combined | | 17.57 | 10,166 | 50.83 | 2.594 |
| Separated | Primary class | 17.55 | 8,654 | 43.27 | 2.616 |
| | Attic cholesteatoma | 17.55 | 11,365.2 | 56.83 | 2.570 |
| | Myringitis | 17.55 | 11,147.2 | 55.74 | 2.558 |
| | Otomycosis | 17.55 | 11,272 | 56.36 | 2.572 |
| | Ventilating tube | 17.55 | 7,087.8 | 35.44 | 2.577 |
| | Sum | | 49,526.2 | 247.64 | 12.893 |

## Discussion

In real practice, it is not easy to examine the status of TM and reach an accurate diagnosis of the middle ear in crying children or non-cooperative patients in a short time. Additionally, in situations where a skilled otologist is not available, there is likely to be an incorrect diagnosis, which leads to malpractice. Although diagnostic rates have dramatically increased since the otoendoscopy was introduced, diagnostic accuracy still differs among physicians [2], while even otolaryngologists can sometimes produce inaccurate diagnoses [3]. Therefore, many researchers have worked on various deep learning models for the effective diagnosis of middle ear diseases.

Previous studies have shown that deep-learning classification can accurately predict the diagnosis of otitis media, up to almost 98.26% of the time [8,9,12]. Alhudhaif et al. [8] analyzed a total 956 otoendoscopic images divided into five classes consisting of otitis externa, ear ventilating tube, foreign bodies in the ear, pseudo-membranes, and tympanosclerosis with an overall accuracy rate of 98.26%. Khan et al. [9] analyzed 2,484 otoendoscopic images divided into three classes consisting of normal, perforation, and middle ear effusion with an overall accuracy rate of 95%. Zeng et al. [12] analyzed 20,542 otoendoscopic images divided into eight classes consisting of normal, cholesteatoma of the middle ear, chronic suppurative otitis media, external auditory canal bleeding, impacted cerumen, otomycosis external, secretory otitis media, and tympanic membrane calcification with an overall accuracy rate of 95.59%. However, these studies were limited by the fact that only one diagnostic label per image was assigned for deep-learning prediction, despite the fact that multiple diseases can be detected simultaneously in real practice. For example, some patients with attic cholesteatoma can have ventilating tube for prevention of TM retraction, while we can also diagnose myringitis in a patient who has tympanic perforation with or without tympanosclerosis.

In this study, we proposed a deep-learning method that can predict the diagnosis of TM changes for two non-coexisting diseases (OME and COM) and four concurrently detectable categories (attic cholesteatoma, myringitis, otomycosis and ventilating tube) with a single network. Our deep-learning classification demonstrated high predictive performance using a database including TMs with up to 4 diseases at the same time. The DSC value of the primary class was greater than 95%, with COM achieving the highest value. In terms of secondary classes, the ventilating tube was rarely misidentified (DSC = 98.89%). Therefore, the multi-class classification for TM changes may have potential for higher clinical applicability than previous approaches in which all images were single labeled.

The combined model for predicting multiple classes at the same time produced better outcomes and required less inference time than the separate models that required a per-class training. The combined model made its prediction by comprehensively observing the entire tympanic membrane (Fig 5). The combined model also finished the prediction in 1/5 of the

training and inference time required for separate models (Table 3). These advantages of deep-learning prediction can help improve the overall diagnostic quality for TM changes. Due to their high predictability, the deep learning models can also support clinical decision-making for inexperienced clinicians and be utilized as a training tool for medical staff. The reduced analysis time of the deep learning models can also make real-time application more feasible. In the same regard, deep learning prediction can help with more accurate diagnoses beyond the constraints of time and space through tele-medicine. Finally, their high reproducibility can enhance the reliability and objectivity of the analysis tool for diagnosis.

However, there were still some limitations on this study. First, even though a large amount of samples were collected for analysis, the deep learning dataset was collected from a single center. Second, a small sample size of otomycosis resulted in fewer training opportunities, thus impairing its predictability. Third, as the number of positives in the secondary classes increases, the number of the secondary classes correctly predicted decreased, even in multi-class classification. An extended dataset with diverse disease patterns can be used to validate the generality and robustness of our classification and improve the prediction performance of TM changes. In the same vein, when applied to otoendoscopic video sequences [15], it can help overcome the bias of still image-based prediction. Cerumen, which was not included in this study, may limit the information on TMs required for diagnosis. As part of the pre-diagnosis evaluation process, quantifying the amount of cerumen using deep-learning segmentation would be helpful to determine whether cleaning of external acoustic meatus is necessary for accurate diagnosis. Ultimately, it is necessary to develop diagnostic tools that anyone can use in the EAC to easily diagnose otologic diseases.

## Conclusions

In the present study, we developed a multi-class classification method for predicting TM changes using deep-learning. The deep-learning algorithm accurately diagnosed the TM changes on otoendoscopic images, even for multiple concurrent diseases. Using the combined model, the inference time per image was reduced to 2.594 ms (more than 380 images can be processed per second), which indicates that deep-learning prediction can be applicable in real-time. Therefore, deep-learning classification can support clinical decision-making by accurately and reproducibly predicting tympanic membrane changes in real time, even in the presence of multiple concurrent diseases.

## Author Contributions

**Conceptualization:** Jihoon Kweon, Joong Ho Ahn.

**Data curation:** Yeonjoo Choi, Jaehee Hur, Joong Ho Ahn.

**Formal analysis:** Jihye Chae, Keunwoo Park, Jaehee Hur, Jihoon Kweon.

**Investigation:** Yeonjoo Choi, Joong Ho Ahn.

**Methodology:** Jihye Chae, Keunwoo Park.

**Project administration:** Yeonjoo Choi, Jihoon Kweon, Joong Ho Ahn.

**Resources:** Yeonjoo Choi, Keunwoo Park, Jaehee Hur.

**Software:** Jihye Chae, Keunwoo Park, Jaehee Hur.

**Supervision:** Jihoon Kweon, Joong Ho Ahn.

**Validation:** Jihoon Kweon.

**Visualization:** Keunwoo Park.

**Writing – original draft:** Yeonjoo Choi, Jihye Chae.

**Writing – review & editing:** Jihoon Kweon, Joong Ho Ahn.

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
