## [Decision Letter · Decision Letter 0]

21 Jun 2022

PONE-D-22-12241Automated Multi-class Classification of Otitis Media using Deep LearningPLOS ONE

Dear Dr. Ahn,

Thank you for submitting your manuscript to PLOS ONE. After careful consideration, we feel that it has merit but does not fully meet PLOS ONE’s publication criteria as it currently stands. Therefore, we invite you to submit a revised version of the manuscript that addresses the points raised during the review process. If the authors choose to submit a revised version of the manuscript, please include an itemized and detailed response to the comments made by the Editor and the Reviewers (see below).

We look forward to receiving your revised manuscript.

Kind regards,

Rafael da Costa Monsanto, M.D.

Academic Editor

PLOS ONE

Journal Requirements:

"This work was supported by the National Research Foundation of Korea(NRF) grant funded by the Korea government(MSIT).(No. 2021R1A2C2010048)"

Additional Editor Comments:

Please address the comments made by all the reviewers. Although reviewers agree this is an interesting study, there are several concerns that must be addressed before the article is considered for publication. Major concerns included:

- Grammar and syntax review is necessary;

- A more detailed description of the methods is needed (calculation of sample size, validation of methods, experience of examiners, inter-observer agreement, etc);

- The IRB protocol number must be included in the body of the manuscript;

- What does the category "None" entitles;

- The lack of inclusion / differentiation of acute otitis media and otomycosis.

Please include an itemized, detailed response to the comments made by all reviewers. If not yet done so, please make all data available or provide a detailed explanation why some of the data must not be shared.

Reviewers' comments:

Reviewer's Responses to Questions

**Comments to the Author**

1. Is the manuscript technically sound, and do the data support the conclusions?

Reviewer #1: Partly

Reviewer #2: Yes

Reviewer #3: Partly

2. Has the statistical analysis been performed appropriately and rigorously? 

Reviewer #1: Yes

Reviewer #2: Yes

Reviewer #3: I Don't Know

3. Have the authors made all data underlying the findings in their manuscript fully available?

Reviewer #1: Yes

Reviewer #2: Yes

Reviewer #3: Yes

4. Is the manuscript presented in an intelligible fashion and written in standard English?

Reviewer #1: No

Reviewer #2: Yes

Reviewer #3: Yes

5. Review Comments to the Author

Reviewer #1: Dear authors

Thank you for this interesting study. This maybe helpful for general practitioners, pediatricians and other specialities that may work with some patients and will present some challenges for their diagnosis of middle ear diseases. However, the presentation of data in tables and figures look even more complete than theie explanations on the manuscript. Also there are some detailed issues with English writing. This manuscript will be totally benefit of a refinement of syntax and grammar. Some phrases look pretty colloquial for the scope of this journal.

There are some other questions regarding the manuscript

1. Why don't you included the acute otitis media? this is one of the most common ear diseases in children and the assessment of this disease will enrich the study.

2. In the study, you only included two researchers, and one otologist that we do not know their time at practice, that would be really helpful to know it. Another question is variability between researchers, did you take it during the study? and also the variability between ears of a same patient, I think, this needs to be addressed. You may need to clarify if both researchers were blinded or not and how was blinding

3. It is a bit tough to know what is first and secondary class, throughout the text, is very confusing for readers

4. The classification 'none', sometimes is confused with OME and COM, this needs to be clear for readers

5. Will be very interesting if the discussion is refined and comparison with other softwares and tools is made

Your work is very interesting but needs to be refined in grammar, syntax and some specific details in the methodology and results

Reviewer #2: Thank you very much for the opportunity to review this quite fascinating manuscript for PLOS ONE. In this retrospective study, authors aimed to assess the impact of concurrent changes of the tympanic membrane (perforation, myringitis and more) by using deep learning. Despite of this interesting approach, I have few comments.

1. TITLE: I think it should me more appropriate if authors change the title for something more related to classification of tympanic membrane changes by using deep learning assessment.

2. MATERIALS AND METHODS: How the samples were collected? What equipment was used and how images were processed? How distant from the tympanic membrane photos were taken? Was the method reproducible?

3. Please, write the IRB number in the manuscript.

4. Was the method already validated anteriorly?

5. It is interesting that ventilation tubes were not confused with TM perforation.

6. I suggest the authors to write a paragraph discussing the importance of the deep learning in assessing the tympanic membrane also in times of telemedicine.

7. Line 199: "images inyo three classes" (into?)

8. I suggest the authors to re-write the last paragraph in order to add the conclusion section.

9. D the authors found any correlation among size of perforation and better results in automated classification?

Reviewer #3: This retrospective study built a database of otoendoscopic images including multiple diseases and investigate the impact of concurrent diseases on the classification performance of deep learning network on the diagnostic performance using algorithm for multi-class classification of otitis media.

However, I would like to point out some aspects that need clarification.

1) For the primary classification of the images in the 3 categories, the category "NONE" was not clear which cases would be included, if images without diagnosis of chronic otitis media and otitis media with effusion, or if there would also be images of normal ears, cerumen? In addition, for the secondary classification, there was the allocation of a very small number of certain cases, mainly of otomycoses, which greatly impaired the accuracy of this diagnosis.

2) As the author himself reports in the justifications for this study, there are several criteria for the diagnosis of tympanic membrane lesions and the criteria used by specialists (a skilled otologist) for the diagnosis of myringitis in this study were not described.

3) It needs to be clarified why the construction of this algorithm does not include the diagnosis of acute otitis media and cerumen, which could certainly impact the otoendoscopies performed by other non-ENT professionals who did not clean the external acoustic meatus.

4) Several deep learning models for the diagnosis of middle ear diseases have already been developed and it is not described what are the real differences of the previous models in relation to the combined model of this study and if the images used were still images or otoendoscopic video sequence?

5) It should be further described how this model can reduce inference time and computational resources for diagnostic support.

6) the titles of figures 2, 3 and 4 are in bold and without focus, making it very difficult to read.

7) The conclusion in the abstract is extremely broad, making inferences that cannot be supported with the result presented and we emphasize that what is described at the end of the manuscript is much more faithful to the results presented in the study.

8) We suggest checking the writing of the manuscript because there are typos.

6. PLOS authors have the option to publish the peer review history of their article (what does this mean?). If published, this will include your full peer review and any attached files.

Reviewer #1: No

Reviewer #2: No

Reviewer #3: No

---

## [Author Response · Author response to Decision Letter 0]

11 Aug 2022

Response to the comments

Reviewer #1: Dear authors

Thank you for this interesting study. This maybe helpful for general practitioners, pediatricians and other specialities that may work with some patients and will present some challenges for their diagnosis of middle ear diseases. However, the presentation of data in tables and figures look even more complete than theie explanations on the manuscript. Also there are some detailed issues with English writing. This manuscript will be totally benefit of a refinement of syntax and grammar. Some phrases look pretty colloquial for the scope of this journal.

<Answer> We thank you for your time and input. We have responded to the comments below.

There are some other questions regarding the manuscript

1. Why don't you included the acute otitis media? this is one of the most common ear diseases in children and the assessment of this disease will enrich the study.

<Answer> We agree that acute otitis media (AOM) is one of the common otologic diseases, especially in children. In our study, AOM falls under the secondary class of ‘myringitis’, defined as any inflammation on tympanic membrane. If the deep-learning model correctly predicts for AOM, the primary and secondary classes will be ‘None’ and ‘myringitis’, respectively. We added a sentence describing the classification setting for AOM. 

Added in page 7:

Myringitis is defined as any inflammation of the tympanic membrane, including acute otitis media.

2. In the study, you only included two researchers, and one otologist that we do not know their time at practice, that would be really helpful to know it. Another question is variability between researchers, did you take it during the study? and also the variability between ears of a same patient, I think, this needs to be addressed. You may need to clarify if both researchers were blinded or not and how was blinding

<Answer> In our study, two experienced otologists participated in the annotation of otoendoscopic images, which did not contain any patient-identifiable information. They blindly assigned labels to each image. When two annotators labeled an image identically, the image was included in the dataset due to 'complete agreement'. We agree that the current description of the labeling process may raise concerns about bias in data selection. The relevant sentences were changed to clarify the labeling process of our study.

Changed in page 7:

The categories of each image were blindly annotated by two otologists with 26 and 5 years of experience, respectively. A total of 6,630 otoendoscopic images labeled identically by two annotators were included in this study.

3. It is a bit tough to know what is first and secondary class, throughout the text, is very confusing for readers

<Answer> To explain the combination of the primary class (OME, COM, 'None') and the secondary classes (attic cholesteatoma, myringitis, otomycosis and ventilating tube), Figure 2(b) was added as suggested.

Changed in Figure 2:

Figure 2

(a) Schematic diagram of deep learning network for multi-class classification of otoendoscopic images. (b) Labeling examples. For a normal tympanic membrane (TM), the otoendoscopic image was labeled as 'None' for the primary class and 'False' for the secondary classes (attic cholesteatoma, myringitis, otomycosis and ventilating tube). When TM was diseased as one of the secondary classes without otitis media with effusion (OME) and chronic otitis media (COM), the primary class was given as 'None' for the otoendoscopic image.

4. The classification 'none', sometimes is confused with OME and COM, this needs to be clear for readers

<Answer>

The difference between previous approaches and our study is 'None' in the primary classes. When TM falls under any of the secondary classes without OME and COM, the primary class was assigned as 'None'. Since the prediction of the secondary class was binary in nature (True/False), the absence of the disease could be predicted. However, the primary class had two options of OME and COM (multi-class classification), so the absence of COM and OME should be included as a class in the primary class. Together with Figure 2(b), the sentences describing the meaning of 'None' were added as suggested. 

Added in pages 7 and 8:

For example, when a TM was normal, the primary class was 'None' and the secondary classes were 'False' for attic cholesteatoma, myringitis, otomycosis, and ventilating tube (Fig. 2b). An otoendoscopic image with only otomycosis was assigned 'None' for the primary class, 'True' for otomycosis, and 'False' for the other secondary classes.

5. Will be very interesting if the discussion is refined and comparison with other softwares and tools is made

<Answer>

We agree that the comparison will be interesting. However, there is currently no open-source or commercial solution for TM diagnosis. We will consider it if/when such a solution is available. The discussion section was reorganized and rephrased as suggested too. Since major revisions were made to the Discussion section, we regret that we cannot provide complete revisions below. 

Your work is very interesting but needs to be refined in grammar, syntax and some specific details in the methodology and results

<Answer>

Grammar, syntax, and other expressions that needed to be improved have been corrected as suggested. We regret that we cannot provide a complete list of revisions due to space issues.

 

Reviewer #2: Thank you very much for the opportunity to review this quite fascinating manuscript for PLOS ONE. In this retrospective study, authors aimed to assess the impact of concurrent changes of the tympanic membrane (perforation, myringitis and more) by using deep learning. Despite of this interesting approach, I have few comments.

<Answer> We thank you for your time and input. We have responded to the comments below.

1. TITLE: I think it should me more appropriate if authors change the title for something more related to classification of tympanic membrane changes by using deep learning assessment.

<Answer> Thank you for your suggestion. If the editorial policy of PLoS One permits the change of the publication title, we will change the title as follows: ' Multi-class Classification for Prediction of Tympanic Membrane Changes With Deep Learning Models'

2. MATERIALS AND METHODS: How the samples were collected? What equipment was used and how images were processed? How distant from the tympanic membrane photos were taken? Was the method reproducible?

<Answer> In the clinical routine practice, otologists performed diagnostic examination using real-time video sequence and captured an image frame visualizing the whole TM for diagnosis. The otoendoscopic images completely anonymized were stored in the hospital system. After the IRB review, the dataset requested was provided to researchers. The imaging devices were listed below. The pre-processing for the collected images were described in Figure 2. The distance of otoendoscopy from the tympanic membrane varied according to the shape of patient's auditory canal. One of the biggest advantages of deep learning is its reproducibility. No differences were found in repeated tests. Information about sample collection and image acquisition condition was added in the manuscript, as suggested.

Endoscopy digital processor

Olympus VISERA ELITE 2

Olympus VISERA CLV-S40

Olympus OTV-SP1

Camera head

Olympus CH-S200-XZ-EB

Olympus OTV-SP1H-NA-12E 

Added in page 7:

In clinical practice, the otoendoscopic video sequence was taken for diagnostic examination and an image frame visualizing the whole TM was stored in the hospital system without patient-identifiable information. Otoendoscopic images enrolled based on the date of visit were completely anonymized before being provided by the hospital system.

3. Please, write the IRB number in the manuscript.

<Answer> IRB number was added in the manuscript as suggested.

Added in Page 7:

The present study is in compliance with the Declaration of Helsinki and research approval was granted from the Institutional Review Board of the Asan Medical Center with a waiver of research consent (IRB no. 2021-0837).

4. Was the method already validated anteriorly?

<Answer> EfficientNet-B4, which was used as the basis of our deep-learning models, is a network architecture that has been validated through a lot of studies. For the customized network of our study for TM change, the performance was validated through an ablation test. 

5. It is interesting that ventilation tubes were not confused with TM perforation.

<Answer> In the prediction of ventilation tube, the deep-learning model in our study failed in only 37 cases (0.56% of entire images) and the mis-identification between the ventilation tube and perforation (COM) was rarely observed (3 cases). Please see the images for the false predictions in attached file.

6. I suggest the authors to write a paragraph discussing the importance of the deep learning in assessing the tympanic membrane also in times of telemedicine.

<Answer> The importance of deep-learning classification in the use of tele-medicine was described in Discussion section. 

Changed in page 14:

The combined model for predicting multiple classes at the same time produced better outcomes and required less inference time than the separate models that required a per-class training. The combined model made its prediction by comprehensively observing the entire tympanic membrane (Fig. 5). The combined model also finished the prediction in 1/5 of the training and inference time required for separate models (Table 3). These advantages of deep-learning prediction can help improve the overall diagnostic quality for TM changes. Due to their high predictability, the deep learning models can also support clinical decision-making for inexperienced clinicians and be utilized as a training tool for medical staff. The reduced analysis time of the deep learning models can also make real-time application more feasible. In the same regard, deep learning prediction can help with more accurate diagnoses beyond the constraints of time and space through tele-medicine. Finally, their high reproducibility can enhance the reliability and objectivity of the analysis tool for diagnosis.

7. Line 199: "images inyo three classes" (into?)

<Answer> The typo was corrected as suggested.

8. I suggest the authors to re-write the last paragraph in order to add the conclusion section.

<Answer>

The last paragraph of Discussion section was reorganized and rephrased as suggested. Following the Editorial policy of PLoS One, a conclusion section was added. 

Added in pages 15 and 16:

Conclusions

In the present study, we developed a multi-class classification method for predicting TM changes using deep-learning. The deep-learning algorithm accurately diagnosed the TM changes on otoendoscopic images, even for multiple concurrent diseases. Using the combined model, the inference time per image was reduced to 2.594 ms (more than 380 images can be processed per second), which indicates that deep-learning prediction can be applicable in real-time. Therefore, deep-learning classification can support clinical decision-making by accurately and reproducibly predicting tympanic membrane changes in real time, even in the presence of multiple concurrent diseases.

9. D the authors found any correlation among size of perforation and better results in automated classification?

<Answer>

For perforations smaller than a quarter of TM area, the sensitivity was 93.68% (474/506), while the overall sensitivity was 95.37% (1,463/1,534). The area analysis was performed visually. In the future work, quantification analysis of perforation and TM areas will be assessed using deep-learning segmentation.

Reviewer #3: This retrospective study built a database of otoendoscopic images including multiple diseases and investigate the impact of concurrent diseases on the classification performance of deep learning network on the diagnostic performance using algorithm for multi-class classification of otitis media.

However, I would like to point out some aspects that need clarification.

<Answer> We thank you for your time and input. We have responded to the comment below.

1) For the primary classification of the images in the 3 categories, the category "NONE" was not clear which cases would be included, if images without diagnosis of chronic otitis media and otitis media with effusion, or if there would also be images of normal ears, cerumen? In addition, for the secondary classification, there was the allocation of a very small number of certain cases, mainly of otomycoses, which greatly impaired the accuracy of this diagnosis.

<Answer> 'None' indicates the absence of OME and COM. The primary class had two options of OME and COM (multi-class classification), so 'None' should be included as a class in the primary class. To explain the combination of the primary class (OME, COM, 'None') and the secondary classes (attic cholesteatoma, myringitis, otomycosis and ventilating tube), Figure 2(b) was added and the relevant sentences were added in page 7 as suggested.

We agree that the small number of otomycosis samples impaired the classification performance. However, due to the low prevalence of the disease, it was not possible to increase the dataset size. This issue was described as a limitation in our study. In the future, an extended dataset collected from multiple centers will help improve the predictability. 

Added in pages 7 and 8:

For example, when a TM was normal, the primary class was 'None' and the secondary classes were 'False' for attic cholesteatoma, myringitis, otomycosis, and ventilating tube (Fig. 2b). An otoendoscopic image with only otomycosis was assigned 'None' for the primary class, 'True' for otomycosis, and 'False' for the other secondary classes.

Changed in Figure 2:

Figure 2

(a) Schematic diagram of deep learning network for multi-class classification of otoendoscopic images. (b) Labeling examples. For a normal tympanic membrane (TM), the otoendoscopic image was labeled as 'None' for the primary class and 'False' for the secondary classes (attic cholesteatoma, myringitis, otomycosis and ventilating tube). When TM was diseased as one of the secondary classes without otitis media with effusion (OME) and chronic otitis media (COM), the primary class was given as 'None' for the otoendoscopic image.

2) As the author himself reports in the justifications for this study, there are several criteria for the diagnosis of tympanic membrane lesions and the criteria used by specialists (a skilled otologist) for the diagnosis of myringitis in this study were not described.

<Answer> The criteria for each class were added as suggested.

Added in page 7:

OME refers to effusions in the middle ear cavity, which manifest in the air-fluid level or as an amber-like color change of TMs. COM refers to a perforated TM.

Attic cholesteatoma refers to any sign of retraction pocket in attic or visible attic destruction. Myringitis is defined as any inflammation of the tympanic membrane, including acute otitis media. Otomycosis refers to a fibrinous accumulation of debris or visible pores of fungus in the external auditory canal. Ventilating tube refers to an inserted tube across the TM.

3) It needs to be clarified why the construction of this algorithm does not include the diagnosis of acute otitis media and cerumen, which could certainly impact the otoendoscopies performed by other non-ENT professionals who did not clean the external acoustic meatus.

<Answer> We agree that acute otitis media (AOM) a rather common otologic disease, especially in children. In our study, AOM falls under the secondary class of ‘myringitis’, defined as any inflammation on tympanic membrane. If the deep-learning model correctly predicts for AOM, the primary and secondary classes will be ‘None’ and ‘myringitis’, respectively. We added a sentence describing the classification setting for AOM. 

Some previous studies included 'cerumen' as a class for deep-learning classification as you described. Since this study only evaluated the diagnostic performance when sufficient information about the tympanic membrane was available, we assumed that the evaluation of cerumen was performed prior to the TM diagnosis. The impact of cerumen on prediction performance will be assessed using deep learning segmentation in our future work. We added a few sentences describing the issue of cerumen.

Added in page 7:

Myringitis is defined as any inflammation of the tympanic membrane, including acute otitis media.

Added in page 15:

Cerumen, which was not included in this study, may limit the information on TMs required for diagnosis. As part of the pre-diagnosis evaluation process, quantifying the amount of cerumen using deep-learning segmentation would be helpful to determine whether cleaning of external acoustic meatus is necessary for accurate diagnosis.

4) Several deep learning models for the diagnosis of middle ear diseases have already been developed and it is not described what are the real differences of the previous models in relation to the combined model of this study and if the images used were still images or otoendoscopic video sequence?

<Answer>

The main difference between previous studies and our method is that our model predicts multiple labels for a single image, while all images were single labeled in previous studies. The combined model also requires less time for deep-learning prediction, allowing real-time application. Our dataset consisted of still images, and the advantage of otoendoscopic video sequences in the deep-learning classification of TM changes was described in Discussion, as shown below. To clarify the difference between previous approaches and our study, the paragraphs below were rephrased and reorganized.

Changed in page 14:

In this study, we proposed a deep-learning method that can predict the diagnosis of TM changes for two non-coexisting diseases (OME and COM) and four concurrently detectable categories (attic cholesteatoma, myringitis, otomycosis and ventilating tube) with a single network. Our deep-learning classification demonstrated high predictive performance using a database including TMs with up to 4 diseases at the same time. The DSC value of the primary class was greater than 95%, with COM achieving the highest value. In terms of secondary classes, the ventilating tube was rarely misidentified (DSC = 98.89%). Therefore, the multi-class classification for TM changes may have potential for higher clinical applicability than previous approaches in which all images were single labeled.

The combined model for predicting multiple classes at the same time produced better outcomes and required less inference time than the separate models that required a per-class training. The combined model made its prediction by comprehensively observing the entire tympanic membrane (Fig. 5). The combined model also finished the prediction in 1/5 of the training and inference time required for separate models (Table 3).

5) It should be further described how this model can reduce inference time and computational resources for diagnostic support.

<Answer> Information about the computational cost and inference time of deep-learning prediction was added in Table 3. Descriptions of the inference time were also added to the Abstract and Discussion section

Added in Abstract: 

The inference time per image was 2.594 ms on average.

Added in Table 3:

Changes in page 14:

The combined model for predicting multiple classes at the same time produced better outcomes and required less inference time than the separate models that required a per-class training. The combined model made its prediction by comprehensively observing the entire tympanic membrane (Fig. 5). The combined model also finished the prediction in 1/5 of the training and inference time required for separate models (Table 3).

6) the titles of figures 2, 3 and 4 are in bold and without focus, making it very difficult to read.

<Answer> Resolutions of figures 2, 3 and 4 were improved (doubled in each axis) as suggested.

7) The conclusion in the abstract is extremely broad, making inferences that cannot be supported with the result presented and we emphasize that what is described at the end of the manuscript is much more faithful to the results presented in the study.

<Answer>

The Conclusion section in the Abstract was rephrased to concisely explain the implication of our study.

Changed in Abstract:

Deep-learning classification can be used to support clinical decision-making by accurately and reproducibly predicting tympanic membrane changes in real time, even in the presence of multiple concurrent diseases.

8) We suggest checking the writing of the manuscript because there are typos.

<Answer>

Grammar, syntax, and other expressions that needed to be improved have been corrected as suggested. We regret that we cannot provide a complete list of revisions due to space issues.

---

## [Decision Letter · Decision Letter 1]

26 Sep 2022

Automated Multi-class Classification for Prediction of Tympanic Membrane Changes with Deep Learning Models

PONE-D-22-12241R1

Dear Dr. Ahn,

We’re pleased to inform you that your manuscript has been judged scientifically suitable for publication and will be formally accepted for publication once it meets all outstanding technical requirements.

Kind regards,

Rafael da Costa Monsanto, M.D.

Academic Editor

PLOS ONE

Additional Editor Comments (optional):

Reviewers' comments:

Reviewer's Responses to Questions

**Comments to the Author**

1. If the authors have adequately addressed your comments raised in a previous round of review and you feel that this manuscript is now acceptable for publication, you may indicate that here to bypass the “Comments to the Author” section, enter your conflict of interest statement in the “Confidential to Editor” section, and submit your "Accept" recommendation.

Reviewer #1: All comments have been addressed

Reviewer #3: All comments have been addressed

2. Is the manuscript technically sound, and do the data support the conclusions?

Reviewer #1: Yes

Reviewer #3: Yes

3. Has the statistical analysis been performed appropriately and rigorously? 

Reviewer #1: Yes

Reviewer #3: Yes

4. Have the authors made all data underlying the findings in their manuscript fully available?

Reviewer #1: Yes

Reviewer #3: Yes

5. Is the manuscript presented in an intelligible fashion and written in standard English?

Reviewer #1: Yes

Reviewer #3: Yes

6. Review Comments to the Author

Reviewer #1: Dear authors

Thank you for addressing all our comments. Now the manuscript looks even clearer and concise. The techniques and the graphics are pretty well explained which highlights the relevance of the methods used.

Reviewer #3: Thank you for thoroughly addressing the comments. The authors have provided corresponding information and the manuscript has improved overall. Findings from the additional analysis are interesting and remain relevant.

7. PLOS authors have the option to publish the peer review history of their article (what does this mean?). If published, this will include your full peer review and any attached files.

Reviewer #1: No

Reviewer #3: No

---

## [Editor Report · Acceptance letter]

28 Sep 2022

PONE-D-22-12241R1 

Automated Multi-class Classification for Prediction of Tympanic Membrane Changes with Deep Learning Models 

Dear Dr. Ahn:

I'm pleased to inform you that your manuscript has been deemed suitable for publication in PLOS ONE. Congratulations! Your manuscript is now with our production department. 

Kind regards, 

on behalf of

Dr. Rafael da Costa Monsanto 

Academic Editor

PLOS ONE